# Establishment of an Antimicrobial Stewardship Program to Spare the Use of Oral Fluoroquinolones for Acute Uncomplicated Cystitis in Outpatients

**DOI:** 10.3390/antibiotics13090886

**Published:** 2024-09-14

**Authors:** Tomoyuki Kato, Masayuki Nagasawa, Ippei Tanaka, Yuka Seyama, Reiko Sekikawa, Shiori Yamada, Eriko Ishikawa, Kento Kitajima

**Affiliations:** 1Department of Pharmacy, Musashino Red Cross Hospital, 1-26-1, Kyonancho, Musashino-shi, Tokyo 180-8610, Japan; t.kato@musashino.jrc.or.jp (T.K.); i.b.d.a.p.9025@gmail.com (I.T.); yu_ka_kirakira@yahoo.co.jp (Y.S.); yreeko@gmail.com (R.S.); s.luckyf2pnd@gmail.com (S.Y.); e.ishikawa0909@gmail.com (E.I.); k.kitajima.mp14@gmail.com (K.K.); 2Department of Infection Control, Musashino Red Cross Hospital, 1-26-1, Kyonancho, Musashino-shi, Tokyo 180-8610, Japan; 3Department of Pediatrics, Musashino Red Cross Hospital, 1-26-1, Kyonancho, Musashino-shi, Tokyo 180-8610, Japan

**Keywords:** antimicrobial stewardship team, antimicrobial stewardship program, acute uncomplicated cystitis, antibiogram, cephalosporin, fluoroquinolone, outpatient setting

## Abstract

The increase in fluoroquinolone (FQ)-resistant *Escherichia coli* (EC) is a serious global problem. In addition, much of acute uncomplicated cystitis (AUC) cases are caused by EC. FQs have been selected for the treatment of cystitis in outpatients, and there is concern about treatment failure. It is therefore necessary to select appropriate antimicrobials to spare FQs. However, there are few reported effective antimicrobial stewardship programs (ASPs) for outpatients. We aimed to establish the effective ASP for outpatients diagnosed with AUC caused by EC, to spare the use of FQs, and to explore optimal oral antimicrobials for AUC. The study subjects were outpatients treated for AUC caused by extended-spectrum β-lactamase-non-producing EC (non-ESBL-EC). Based on the antibiogram results, we recommended cefaclor (CCL) as the initial treatment for AUC, and educated clinical pharmacists who also worked together to advocate for CCL or cephalexin (CEX) prescriptions. FQ usages decreased, and cephalosporin (Ceph) prescriptions increased in all medical departments. The Ceph group (n = 114; CCL = 60, CEX = 54) in the non-FQ group had fewer treatment failures than the FQ group (n = 86) (12.3% vs. 31.4%). Cephs, including CCL and CEX, were effective treatments for AUC caused by non-ESBL-EC. Antimicrobial selection based on antibiogram results and the practice of an ASP in collaboration with clinical pharmacists were useful for optimizing antimicrobial therapy in outpatients.

## 1. Introduction

Quinolones are bactericidal antimicrobials that act on DNA gyrase and topoisomerase IV, inhibiting DNA synthesis [1]. The first quinolone to be synthesized, nalidixic acid, is active only against Gram-negative bacilli and is unstable during metabolic processes. Fluoroquinolones (FQs), which have fluorine introduced into the basic quinolone structure, have demonstrated enhanced antibacterial activity against Gram-negative bacteria. A new generation of FQs, levofloxacin (LVFX), was then developed, and these also show antibacterial activity against Gram-positive bacteria. LVFX can be administered orally once a day and is characterized by excellent pharmacokinetics and high bioavailability [2]. In general, FQs also exhibit antibacterial activity against *Pseudomonas aeruginosa*, while sitafloxacin acts on both aerobic and anaerobic bacteria [3]. Therefore, the necessity of using FQs, which have a broad antibacterial spectrum, should be carefully considered. Antimicrobial resistance (AMR) is a threat to humans [4], and resistance to FQs in *Escherichia coli* (EC) is a serious global problem [5]. In Japan, an AMR action plan was formulated in 2015 by the Ministry of Health, Labor, and Welfare, with the goal of reducing the FQ resistance rate of EC to 25% or less [6]. Despite the implementation of antimicrobial stewardship, the FQ resistance rate in EC was 35% in 2020. Based on this result, the second AMR action plan was released in 2023, with the goal of reducing the FQ resistance rate to 30% or less by 2027 [7]. EC is a microorganism that causes a variety of infections, including urinary tract infections (UTIs) and intra-abdominal infections. UTIs are some of the most common bacterial infections in all age groups [8]. In addition, 75–95% of acute uncomplicated cystitis (AUC) and pyelonephritis cases are caused by EC, which means that it is the most commonly detected microorganism in UTI cases [9,10]. FQs have been selected for the treatment of cystitis in outpatients due to their antimicrobial spectrum as well as convenience for administration, such as oral compliance [11], but given the current resistance rate in EC, there is concern about treatment failure.

In Japan, about 90% of total antimicrobial consumption were oral antimicrobials prescribed in outpatient settings [12], but there were few reports on antimicrobial stewardship programs (ASPs) for outpatients [13]. In addition, nitrofurantoin, which is recommended in the guidelines [9], is not available, and sulfamethoxazole-trimethoprim (SMX-TMP) is not covered by insurance for AUC in Japan. Consequently, the selection of effective antimicrobials and the establishment of an ASP are issues of urgency. Therefore, we investigated the selection of antimicrobials for the treatment of cystitis caused by extended-spectrum β-lactamase-non-producing EC (non-ESBL-EC) in clinical practice, particularly in an outpatient setting, and examined the trends in the FQ resistance rate of EC. In our hospital, the antimicrobial stewardship team (AST) collaborated with clinical pharmacists to propose an optimal antimicrobial strategy for AUC. Therefore, the objective of this study was to investigate trends in oral antimicrobial use and to analyze the treatment outcomes in AUC caused by non-ESBL-EC. Then, we aimed to identify optimal antimicrobials based on antimicrobial susceptibility results, and to establish an effective ASP by AST and clinical pharmacists in outpatient settings.

## 2. Results

### 2.1. Study Patients

During the study period, 2802 patients were diagnosed with cystitis in an outpatient setting, and of these, 426 had only EC detected in the urine culture after the date of diagnosis. In 35 of these patients (8.2%), the detected EC was an extended-spectrum β-lactamase (ESBL)-producing strain. Of the remaining 391 patients, 305 were prescribed antimicrobials (Figure 1). These patients were then divided into an FQ group (106 patients) and a non-FQ group (199 patients). Blood cultures were performed in 23.9% of cases, all of which were negative, and no patients had bacteremia. There were more women in the non-FQ group than in the FQ group, and the duration of antimicrobial administration was significantly longer in the non-FQ group than in the FQ group. The women were divided into two age groups, ≤49 years and ≥50 years, in each group, but no relationship was found between the type of antimicrobial selected and age group (Table 1).

### 2.2. Antibiogram Trends for EC

The trends in antimicrobial sensitivity of non-ESBL-EC were evaluated using antibiograms. There was no significant difference between the number of non-ESBL-EC strains detected each year. Although no significant changes in susceptibility were observed for LVFX, cefaclor (CCL), or amoxicillin/clavulanic acid (AMPC/CVA), the susceptibility rate to LVFX appeared to be decreasing. However, the data suggested that the susceptibility rate of non-ESBL-EC to CCL remained high (Table 2).

### 2.3. Trends in Antimicrobial Usage by Each Clinical Department

We calculated antimicrobial usage in clinical departments as prescriptions per 1000 visits. In the Urology Department, the use of FQs decreased significantly over the study period. Although the use of FQs in other departments also decreased, this reduction was not significant (Table 3).

### 2.4. Changes in Antimicrobial Treatment for AUC

There was a significant downward trend in the number of FQ prescriptions for AUC over time. In contrast, the number of prescriptions for cephalosporins (Cephs) for the treatment of AUC increased significantly over time. Of the 40 cases of Cephs in 2023, 14 cases were changed from FQ after clinical pharmacists made prescription suggestions to physicians. On the other hand, in the FQ group, there was only one case in which a valid reason for prescribing FQ was clear, and that was a patient with a known allergy to β-lactam antimicrobials. The prescription of penicillin for AUC decreased over time, but not significantly. For all of these antimicrobials, the duration of administration became shorter (Table 4). Of the other antimicrobials, SMX-TMP was the most prescribed, but no changes were observed in the number of prescriptions or the duration of administration.

### 2.5. Support and Intervention for Antimicrobial Treatment for AUC through Cooperation between AST and Clinical Pharmacists

We predicted the number of prescriptions for 2023 based on the trends in the number of antimicrobial prescriptions for AUC from 2019 to 2022. We then compared this prediction with the actual number of prescriptions. We predicted that the number of FQ prescriptions would remain unchanged from the fourth quarter of 2022 onward, but in reality, it decreased further in 2023. In contrast, the number of prescriptions for other antimicrobials was significantly higher than the predicted number of prescriptions (Figure 2).

### 2.6. Treatment Results of Ceph and FQ for AUC

Of the 305 patients included in this study, the treatment outcome of each antimicrobial was available for 247 patients (81.0%) with Ceph (n = 114), FQ (n = 86) and the others (n = 47). Ceph included CCL and cephalexin (CEX) but no oral third-generation Ceph. All FQs were LVFX. CEX is a first-generation Ceph and was judged to be equivalent to CCL. As for the results of each treatment, 100 patients (87.7%) in the Ceph group were effective, while 59 patients (68.6%) in the FQ group were effective. Multivariable logistics regression was performed (Table 5). Selection of Ceph decreased the odds of antimicrobial treatment failure compared to selection of FQ (odds ratio, 0.28; 95% CI [0.088–0.882]; *p* = 0.030). On the other hand, resistance to CCL increased the odds (odds ratio, 5.01; 95% CI [1.600–15.70]; *p* < 0.01), whereas resistance to LVFX did not increase the risk of antimicrobial failure (*p* = 0.286).

## 3. Discussion

The increasing FQ resistance rate in EC is a serious global problem. Similarly, the increase in the proportion of ESBL-EC is an urgent issue that cannot be ignored. In a large study in Japan, women (both pre- and postmenopausal) had the highest rates of FQ resistance and ESBL-EC [14]. The FQ resistance mechanisms of EC are complex, but they are linked to mutations in the genes encoding DNA gyrase and topoisomerase IV, which are the pharmacological action points of FQs. In addition, decreased drug permeability within bacterial cells due to decreased porin protein expression and increased drug efflux out of the cell are involved, respectively [15]. ESBL is an enzyme that degrades many antimicrobials that have a β-lactam ring, including penicillins and Cephs. The β-lactamase gene encoded on this plasmid is transmitted by species such as EC and *Klebsiella pneumoniae* to other types of bacteria. ESBL consists of three main genotype groups: TEM, SHV, and CTX-M [16]. It has been reported that the CTX-M type is more frequently detected in Japan [17].

The use of antimicrobial agents promotes the development of drug resistance. This is equally applicable to the treatment of infections caused by drug-resistant EC bacteria [18,19]. FQs, including LVFX, are one of the few types of oral antimicrobials that are effective against *Pseudomonas aeruginosa*, and they are therefore highly valuable. In this study, LVFX was most frequently selected because of its high bioavailability [2]. Once-daily dosing is the standard regimen for LVFX and was likely to be preferred by both physicians and patients. Additionally, some FQs (e.g., CPFX) have been discontinued, and are difficult to obtain in Japan.

Furthermore, FQs are important therapeutic agents against infections caused by not only ESBL but also by AmpC β-lactamase-producing Enterobacterales, carbapenem-resistant bacteria, and *Stenotrophomonas maltophilia* [20,21]. There is thus a need to promote behavioral change among physicians, such as discontinuing unnecessary prescription of FQs and instead choosing more appropriate antimicrobials.

AUC is an infectious disease that can affect people of all ages, but the causative bacteria vary according to age and gender. Drug-resistant EC is a common causative agent, especially in postmenopausal women [14]. Therefore, FQs may be recommended for the treatment of AUC in women in the age group that is at risk of infection with these drug-resistant bacteria. Although most patients in this study were women, we detected no relationship between age and antimicrobial selection. This may be because the AST has traditionally practiced prospective audit and feedback (PAF) for the appropriate use of antimicrobials in our hospital. The AST requires preauthorization for intravenous FQs and oral third-generation Cephs [22]. In addition, in terms of the appropriate use of antimicrobials for acute diarrhea and upper respiratory tract inflammation in outpatient settings, the AST monitors the usage status of FQs and macrolides and provides antimicrobial treatment and selection according to the relevant guidance [23].

In our hospital, while these interventions and supports were in place, about half of the FQ prescriptions were administered for seven days or less. While FQs may require long-term administration for infections such as nontuberculous mycobacterial infections, they are generally prescribed for a relatively short period (14 days or less). We had assumed that UTIs would be the most common infectious disease for which physicians selected FQs for short-term prescriptions. This assumption was supported by the fact that urologists were the physicians who most frequently diagnosed AUC and prescribed FQs to outpatients in our hospital. In addition, in this study, male patients accounted for 50% of the FQ group. Although it was not clear which factors contributed to the large number of male patients diagnosed with AUC, it was possible that there was prescription bias or inappropriate antimicrobial use, as most male patients in the FQ group were prescribed by the Urology Department.

After CCL or CEX was recommended for the initial treatment of AUC due to EC, as part of a joint intervention by the AST and clinical pharmacists, the prescription of FQs in the Urology Department decreased significantly during 2023. Additionally, the usage of antimicrobials other than FQs increased. There were no significant changes in other departments. Regarding the types of antimicrobials prescribed, there were significant changes in the number of prescriptions for FQs and Cephs, with the use of FQs decreasing and the use of Cephs increasing. Additionally, the duration of FQ use tended to get shorter, in accordance with the recommended guidelines; this is probably the result of many physicians responding to recommendations from the AST and proposals from clinical pharmacists. Furthermore, there were differences in the actual numbers of antimicrobial prescriptions for AUC treatment in 2023 from the numbers we predicted based on the data for 2019–2022: the number of FQ prescriptions was lower than predicted, while the number of non-FQ antimicrobial prescriptions (mainly CCL) was higher.

Since no other initiatives related to appropriate antimicrobial use were implemented in 2023, the results of this study demonstrate the effectiveness of the intervention of the AST and the prescription support provided by clinical pharmacists. A previous study has shown the usefulness of suppressing the reporting of drug sensitivity results to decrease FQ prescriptions in inpatients [24]. On the other hand, this study could have made physicians aware of the criticality of the FQ resistance problem by clarifying antimicrobial susceptibility. The significance of this study is that we recommended that physicians use Cephs, sparing FQ prescriptions for AUC in outpatient settings, and they agreed with this recommendation.

Although the decline in the prescription of penicillin drugs was not significant, both the amount used and the duration of administration did appear to be decreasing. This is a desirable outcome, according to the antibiogram results. Among non-ESBL-EC, the proportion of strains susceptible to AMPC/CVA remains at the 70% level, making this difficult to recommend as a treatment. However, for CCL, the percentage of susceptible EC strains exceeds 90%, and, to date, the percentage of susceptible strains has not decreased over time. Therefore, in our hospital, CCL is regarded as a narrow-spectrum oral antimicrobial that is effective against AUC caused by EC. In previous reports, the target patients for AUC were female [14], whereas the successful treatment cases in this study included male patients. We could not exclude the possibility that some male patients had complicated pyelonephritis or acute prostatitis. Nevertheless, since cephalosporins may also penetrate the prostate in patients with acute prostatitis, the results of this study indicate a possible switch from intravenous to oral antimicrobial agents. In addition, treatment with Ceph was effective for AUC caused by EC without drug resistance to CCL. Although the surveillance did not include the bacteriological effects of CEX, the results of this study suggest that the efficacy of CEX for non-ESBL-EC is expected.

It is not recommended to use FQ for the treatment of AUC with EC, especially in cases of resistance to LVFX. This result supports the usefulness of selecting effective antimicrobials based on the antibiogram. On the other hand, antimicrobial treatment is not necessarily required for and contributes to the outcome of asymptomatic bacteriuria (ASB) [25]. Furthermore, non-antibiotic therapy is also expected to be effective in treating UTIs in some cases [26].

A limitation of this study was not being able to investigate antimicrobial susceptibility to CEX. In addition, we did not consider the detection of microorganisms other than EC (e.g., *Pseudomonas aeruginosa*). During the study period, among the microorganisms detected in all urine samples at our hospital, including patients with ASB and complicated UTIs, EC accounted for just over half (56.5%). The results for other microorganisms were *Klebsiella pneumoniae* (10.7%), *Enterococcus faecalis* (3.3%), *Proteus mirabilis* (2.7%), and *Pseudomonas aeruginosa* (2.6%). The effects of treatment on AUC in outpatients with *Pseudomonas aeruginosa* detected in their urine culture had not been evaluated. Among the detected ECs, ESBL accounted for 8.2%, but among other microorganisms, the proportions of ESBL- and AmpC-β-lactamase-producing bacteria were 6.2% and 1.7%, respectively. In this study, the proportion of EC and *Klebsiella pneumoniae* detected and the proportion of ESBL-EC showed similar trends as in the surveillance. Considering the detection rate of *Pseudomonas aeruginosa* and drug-resistant bacteria and the results of antibiograms in our hospital, CCL is appropriate as the initial treatment for AUC, but in cases of AUC recurrence or treatment failure after CCL administration, FQs remain appropriate antimicrobials [9,27].

In this study, among treatment failure cases in the FQ group, LVFX resistance was 33.3% and cefaclor resistance was 22.2% in individual patients. Conversely, among the successfully treated cases in the FQ group, LVFX resistance was 16.9%. The appropriateness of the dosage was evaluated based on creatinine clearance by the pharmacist in all cases, and it was found to be appropriate with no excess or deficiency, so the dosage of the antimicrobial used did not seem to be a factor. The possibility remains that FQ resistance may be a factor in affecting treatment outcomes.

Next, no improvement was observed in the detection rate of FQ-resistant EC, and it was not possible to assess the amount of antimicrobial use based on indication. As the quantity of FQs used decreases, the number of susceptible strains is expected to increase, but the actual FQ sensitivity rate, as indicated by the antibiograms, is on the decline. The fact that LVFX resistance did not contribute to antimicrobial treatment outcomes indicates the difficulty in diagnosing UTIs. Thus, this study may have included patients who did not require antimicrobial therapy, such as ASB or cases that resolved spontaneously.

The Musashino Red Cross Hospital is a tertiary emergency medical facility in the North Tama area of Tokyo, Japan. It has more than 10,000 annual emergency transfers, and approximately 1800 outpatients per day. Therefore, outpatients who visit our hospital come from various regions. The detection status of drug-resistant bacteria in the community may be affected by the amount of antimicrobials used in each region. Therefore, we inferred that the effect of reducing FQ usage in our hospital could not be confirmed in outpatient settings.

Although there was no significant difference in the actual number of antimicrobial prescriptions in 2023 compared to the predicted number, extending the study period may help to clarify the relationship between antimicrobial selection, the number of antimicrobial prescriptions, and drug susceptibility results, as well as identify effective support strategies by the AST and clinical pharmacists. Various factors affect the drug susceptibility of microorganisms, not just antimicrobials, but PAF is also needed to avoid unnecessary antimicrobial prescriptions, especially in cases of ASB. In addition, if antimicrobial treatment is necessary, support is needed to ensure that treatment is administered over an appropriate duration.

The World Health Organization proposed the AWaRe categorization, a new method for determining the appropriate use of antimicrobials based on the amount of antimicrobial usages [28]. This classifies antimicrobials into access, watch, and reserve based on priority of use, concerns about AMR, and ease of obtaining antimicrobials. FQs are categorized as watch antimicrobials, which are only prescribed for a specific, limited number of infections; support and interventions should be established to spare the use of FQs. In addition, the efficacy of β-lactam antimicrobials for AUC, including CCL and CEX, is not sufficient in the guidelines [9]. Therefore, it would be valuable to indicate the usefulness of Cephs (especially CEX), as an access antimicrobial agent, for AUC.

## 4. Materials and Methods

### 4.1. Study Design and Setting

The research period extended from 1 January 2019 to 31 December 2023. Data such as patient information were obtained from a medical information data warehouse. This is a database that stores data collected through multiple systems in chronological order. From an infection control perspective, both antimicrobial and infectious disease trends can be monitored, and related data can be compiled [29].

The study participants were outpatients diagnosed with AUC, acute cystitis, or bacterial cystitis, with only EC detected in the urine culture. AUC was defined as an uncomplicated urinary tract infection with one or more of the following symptoms: frequent urination, painful urination, or cloudy urine. All eligible cases were outpatients. Patients who were determined to require inpatient treatment were then excluded. In addition, male patients diagnosed with complicated urinary tract infections were excluded. The culture specimen to be studied was midstream urine or catheter urine, and the urine bacterial count had to be ≥10^5^ CFU/mL before antimicrobial prescription [30]. Patients included adults, children, and infants. The research items were the number of prescriptions including oral antimicrobials, the type of antimicrobial prescribed, the department in which they were prescribed, and the period of administration for each antimicrobial, and each item was tabulated retrospectively. Patients were then divided into FQ prescription group and non-FQ prescription group, and research items were compared in each group. To investigate prescription trends, we also tallied the number of prescriptions and days of administration for each FQ prescribed (Appendix A).

### 4.2. Performing Antimicrobial Susceptibility Tests and Creating Antibiograms

Antibiograms were created to evaluate annual antimicrobial susceptibility results for EC. We evaluated the antimicrobial susceptibility rates of LVFX, CCL, and AMPC/CVA against EC based on the antibiogram preparation guidelines [31,32]. Microscan Neg series (Beckman Coulter, Tokyo, Japan) was used for antimicrobial susceptibility testing. Minimum inhibitory concentrations were determined using the broth microdilution method, and categories of susceptibility (i.e., susceptible, intermediate, or resistant) to antimicrobials were determined according to the Clinical and Laboratory Standards Institute (CLSI) M100-S32. The susceptibility rate was then calculated as a percentage (the number of susceptible bacteria/the number of detected bacteria, multiplied by 100). In addition, among the detected EC, the detection rate of ESBL-producing bacteria was calculated.

### 4.3. Antimicrobial Usage by Clinical Department

After aggregating the number of prescriptions containing antimicrobials for each department, the Urology Department emerged as the one that used FQs the most at our hospital (Appendix A). Therefore, we evaluated FQ antimicrobial usage in this department and others separately. We evaluated outpatient antimicrobial usage as the number of prescriptions per 1000 visits. The number and type of antimicrobials available at our hospital did not change during the study period.

### 4.4. Antimicrobials Recommended for AUC Caused by EC

From January 2019 to December 2022, the antimicrobials to be prescribed for AUC were selected based on the latest antibiogram results for each year for EC in our hospital, together with the guidelines for UTIs from The Japanese Association for Infectious Diseases and the Japanese Society of Chemotherapy [27]. In 2023, as part of antimicrobial stewardship for prescribing physicians, our AST recommended that CCL be preferred as the initial empirical treatment for outpatient AUC.

### 4.5. Collaborative Support System between Clinical Pharmacists and AST

In addition to presenting recommended antimicrobials for AUC in outpatients, the AST presented all healthcare workers at our hospital with the information that the FQ resistance rate in EC, as indicated in the AMR action plan, showed poor improvement over time. Additionally, AST pharmacists educated clinical pharmacists about the bacteria that cause AUC and the antimicrobials for its treatment. After these new initiatives by the AST, clinical pharmacists began, in 2023, to propose CCL as an empirical initial treatment for AUC. Furthermore, when FQ was prescribed as the initial antimicrobial treatment for AUC, the clinical pharmacist questioned the prescribing physician about the reason for choosing it. If a clear reason for selecting FQ could not be obtained from the prescribing physician, the clinical pharmacist then proposed CCL to the prescribing physician as the first-choice antimicrobial and accepted the use of CEX as the second-line drug.

### 4.6. Evaluation of the Therapeutic Efficacy of Ceph and FQ

We retrospectively reviewed medical records to track the course of treatment for AUC from the first antibiotic administration. The effectiveness of treatment was assessed by dividing the patients into the Ceph group and the FQ group. Effective treatment was defined as improvement in bladder irritation, improvement in pyuria, no recurrence of symptoms within 14 days with no change in antimicrobial therapy.

### 4.7. Statistical Analysis

Statistical analyses were carried out using JMP 14 (SAS Institute, Cary, NC, USA). Fisher’s exact test and the Mann–Whitney *U* test were used to compare patient characteristics between the FQ administration group and the non-FQ antimicrobial administration group. Changes over time in the number of antimicrobial prescriptions and antimicrobial susceptibility were evaluated using Spearman’s rank correlation coefficient. Antimicrobial usage in each department was assessed using the number of prescriptions per 1000 outpatient visits. We used exponential smoothing, a time series analysis, to evaluate the effects of these series of AST interventions, including support for prescribing physicians, education for clinical pharmacists, and prescription suggestions from clinical pharmacists. The outcome of treatment was assessed using Fisher’s exact test, and multivariable logistic regression analysis was used to evaluate factors related to treatment outcomes. *p* < 0.05 was considered statistically significant.

## 5. Conclusions

In conclusion, the prescription of antimicrobials for outpatients is a major issue currently and going forward, but antimicrobial selection based on antibiograms according to the institution is useful for optimizing the initial treatment of AUC caused by non-ESBL-EC in outpatients. In addition, a cooperative support system between AST and clinical pharmacists is useful as a method of providing feedback to physicians on optimal antimicrobials. On the other hand, further studies are needed to investigate and establish intervention strategies for patients who do not require antimicrobial therapy in AUC.

## Figures and Tables

**Figure 1 antibiotics-13-00886-f001:**
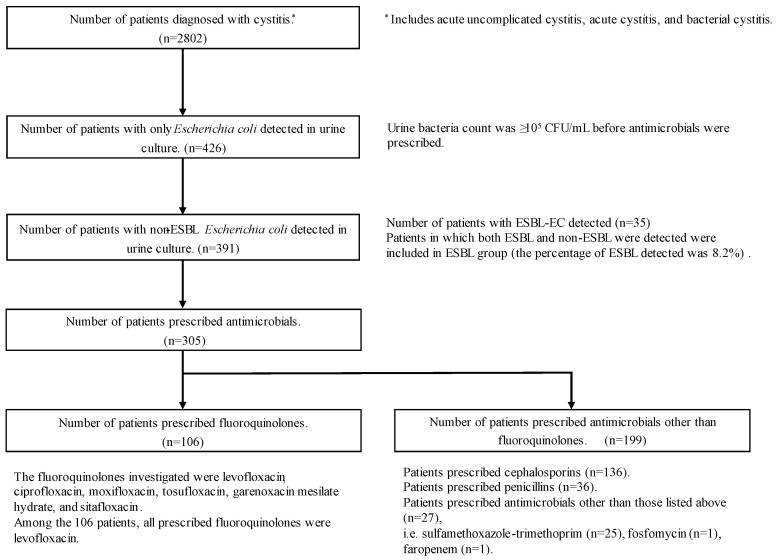
Flow chart of patient enrollment in this study. Abbreviations: ESBL, extended-spectrum β-lactamase; EC, *Escherichia coli.*

**Figure 2 antibiotics-13-00886-f002:**
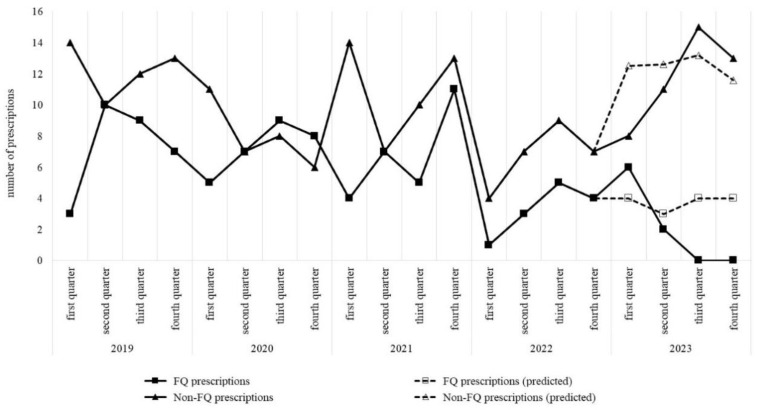
Trends in the number of antimicrobial prescriptions for acute uncomplicated cystitis. The predictiveness evaluation of time series analysis is as follows. Mean absolute error (MAE): 0.286, root mean squared error (RMSE): 0.419. Regarding the number of prescriptions in 2023, the comparison between the actual value group and the predicted value group was evaluated using Mann–Whitney *U* test (respective *p*-values are as follows: 0.103 for FQ prescriptions, <0.05 for non-FQ prescriptions). Abbreviations: FQ, fluoroquinolones.

**Table 1 antibiotics-13-00886-t001:** Patient characteristics in the fluoroquinolones group and other antimicrobials group.

	Fluoroquinolones (n = 106)	Other Antimicrobials ^a^ (n = 199)	*p*-Value
Female (n, %)	53 (50%)	152 (76.4%)	<0.01 ^b^
Year (median, IQR)	66 (49–79)	67 (47–77)	0.424
≤49 years	14	47	0.245
≥50 years	39	105	
Antimicrobials duration (median, IQR)	3 (3–5)	5 (3–7)	<0.01 ^b^

^a^ Includes cephalexin, cefaclor, amoxicillin, amoxicillin/clavulanic acid, sulfamethoxazole-trimethoprim, fosfomycin, and faropenem. ^b^ Statistically significant. Abbreviations: IQR, interquartile range.

**Table 2 antibiotics-13-00886-t002:** Trends in antibiograms related to non-ESBL-EC.

	2019	2020	2021	2022	2023	*p*-Value
Number of detected strains	867	800	786	923	994	0.35
Antimicrobials						
LVFX	83.5	81.9	81.9	80.2	79.4	0.162
CCL	91.2	92.1	92.6	94.1	93.3	0.072
AMPC/CVA	73.6	73.8	73.7	76.2	76.9	0.072

Abbreviations: LVFX, levofloxacin; CCL, cefaclor; AMPC/CVA, amoxicillin/clavulanic acid; ESBL, extended-spectrum β-lactamase; EC, *Escherichia coli*.

**Table 3 antibiotics-13-00886-t003:** Prescriptions per 1000 visits in the Urology Department and other departments.

	2019	2020	2021	2022	2023	Correlation Coefficient ^a^	*p*-Value
Urology							
FQ group	1.690	1.860	1.841	0.948	0.541	−0.363	<0.01 ^b^
Non-FQ group	1.623	1.085	1.761	0.862	1.805	0.064	0.626
Departments other than urology							
FQ group	0.013	0.018	0.014	0.007	0.007	−0.087	0.507
Non-FQ group	0.081	0.064	0.075	0.061	0.100	0.077	0.56

^a^ For correlations between prescriptions per 1000 visits and year of prescription. ^b^ Statistically significant. Abbreviations: FQ, fluoroquinolones.

**Table 4 antibiotics-13-00886-t004:** Trends in the numbers of prescriptions and administration days for each antimicrobial for acute uncomplicated cystitis.

	2019 (n = 78)	2020 (n = 61)	2021 (n = 71)	2022 (n = 40)	2023 (n = 55)	Correlation Coefficient ^c^	*p*-Value
Fluoroquinolones							
Number of prescriptions	29	29	27	13	8	−0.157	<0.01 ^d^
Days of administration (median, IQR)	3 (3–7)	4 (3–7)	5 (3–7)	3 (3–7)	3 (3–5)	−0.039	0.690
Cephalosporins							
Number of prescriptions	33	21	28	14	40	0.156	<0.01 ^d^
Days of administration (median, IQR)	7 (5–7)	7 (3–7)	7 (5–7)	7 (5–7)	5 (3–7)	−0.149	0.084
Penicillin							
Number of prescriptions	10	6	11	6	3	−0.037	0.515
Days of administration (median, IQR)	7 (5–7)	3 (2–5)	6 (3–7)	4 (3–5)	5 (5–7)	−0.131	0.446
Other antimicrobials							
Number of prescriptions	6	5	5	7	4	0.034	0.557
Days of administration (median, IQR)	3 (3–5)	3 (3–14)	5 (3–5)	5 (3–6)	6 (4–7)	0.186	0.352

^c^ For correlations between number of antimicrobial prescriptions and year of prescription. ^d^ Statistically significant. Abbreviations: IQR, interquartile range.

**Table 5 antibiotics-13-00886-t005:** Evaluation of treatment and multivariable logistics regression model for treatment ineffective in acute uncomplicated cystitis.

	Evaluation of Treatment		
Antimicrobials	Ineffective	Effective	*p*-Value
Cephs	14	100	<0.01 ^e^
FQs	27	59	
Variables	Odds ratio (95% CI)		*p*-Value
Antimicrobials			
FQs	Reference		
CCL CEX	0.28 (0.088–0.882) 0.33 (0.104–1.070)		0.030 ^e^ 0.066
Resistant to CCL	5.01 (1.600–15.70)		<0.01 ^e^
Resistant to LVFX	1.64 (0.660–4.090)		0.286
Duration (day)	0.95 (0.815–1.110)		0.488
Age	1.01 (0.985–1.030)		0.625
Sex			
Male	Reference		
Female	0.51 (0.215–1.190)		0.120
Clinical department			
other than urology	Reference		
Urology	1.03 (0.384–2.760)		0.953

^e^ Statistically significant. Cephalosporins included cefaclor (n = 60) and cephalexin (n = 54). Abbreviations: Cephs, cephalosporins; FQs, fluoroquinolones; CI, confidence interval; CCL, cefaclor; CEX, cephalexin; LVFX, levofloxacin.

## Data Availability

The original contributions presented in the study are included in the article/Appendix A, further inquiries can be directed to the corresponding author.

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
