# Peer review of "Establishment of an Antimicrobial Stewardship Program to Spare the Use of Oral Fluoroquinolones for Acute Uncomplicated Cystitis in Outpatients"

_antibiotics, 2024, doi:10.3390/antibiotics13090886_

Round 1

Reviewer 1 Report

Comments and Suggestions for Authors

The authors presented a study on the implementation of an antimicrobial stewardship program to improve antibiotic selection and reduce the use of FQ in patients with AUC. It is useful to see that such an intervention can help to limit the use of FQ, particularly in the urology department. However, the manuscript may still benefit from some revisions. 

Comments.

".... These patients were then divided into an FQ group (106 patients) 82 and a non-FQ group (199 patients), but none of them had bacteremia ..." Did the investigators perform blood cultures in patients with AUC? If not, this statement needs to be rephrased. 

It was interesting to see that almost 50% of the patients were male. Is there any information about the high rate of AUC in male patients (some experts consider UTI in male patients complicated only because of gender)? I did not see a definition for AUC in the methods section. 

Overall antibiotic consumption was evaluated in the study, indication-based consumption will be more useful. If this is not possible, it should be added as a limitation of this study. 

It didn't make sense why FQ treatment failure was more frequent compared to first-generation cephalosporines? Could you provide some more details?  For example; FQ resistance rate by individual patient instead of resistance trends? Dosing? ... 

The trend of FQ and cephalosporin prescribing was quite similar until 2023 (Table 4). A more detailed discussion for this particular year would make the manuscript more interesting for readers. For example, how often did physicians change their prescriptions according to the clinical pharmacists' recommendations?   How many of the FQ prescriptions were without justification? .....

There are several studies that have presented different interventions as part of AMS to reduce the use of FQ in AUC. It would be useful to include these studies in the discussion with a comparison to the current study. Some examples;

- Hayden DA, White BP, Neely S, Bennett KK. Impact of fluoroquinolones

Author Response

Comments 1: Did the investigators perform blood cultures in patients with AUC? If not, this statement needs to be rephrased.

Response 1: Thank you very much for your valuable comment. We agree with this point. Blood cultures were performed in 23.9% of cases, all of which were negative, and none of the patients had bacteremia. We added this to the results section (lines 83-84). Please kindly check it.

Comments 2: Is there any information about the high rate of AUC in male patients (some experts consider UTI in male patients complicated only because of gender)? I did not see a definition for AUC in the methods section.

Response 2: Thank you very much for your valuable comment. We agree with this point. First, we added the definition of AUC in this study to the Materials and Methods section (lines 324 to 328). Next, it was not clear what factors contributed to the large number of male patients diagnosed with AUC. On the other hand, most of the male patients in the FQ group were prescribed by Urology Department, we added a note about the possibility of prescription bias or inappropriate use of antibiotics (lines 212 to 216). Please kindly check it.

Comments 3: Overall antibiotic consumption was evaluated in the study, indication-based consumption will be more useful. If this is not possible, it should be added as a limitation of this study.

Response 3: Thank you very much for your valuable comment. We agree with this point. It was not possible to assess antimicrobial usage based on indication. We added that to the limitation on lines 284-285. Please kindly check it.

Comments 4: It didn't make sense why FQ treatment failure was more frequent compared to first-generation cephalosporines? Could you provide some more details?  For example; FQ resistance rate by individual patient instead of resistance trends? Dosing?

Response 4: Thank you very much for your valuable comment. We agree with this point. The dosage of LVFX was checked by a pharmacist for appropriate dosage in all cases. Therefore, other factors may have contributed to the treatment failure in the FQ group. Drug resistance may affect the treatment outcome of individual patients. We added that to the limitation on lines 277-283. Please kindly check it.

Comments 5: The trend of FQ and cephalosporin prescribing was quite similar until 2023 (Table 4). A more detailed discussion for this particular year would make the manuscript more interesting for readers. For example, how often did physicians change their prescriptions according to the clinical pharmacists' recommendations? How many of the FQ prescriptions were without justification?

Response 5: Thank you very much for your valuable comment. We agree with this point. Of the 40 cephalosporin cases in 2023, 14 were changed from FQ due to a prescription suggestion by a clinical pharmacist, and only one case had a clear justification for choosing FQ. We added that to the limitation on lines 118-122. Please kindly check it.

Comments 6: There are several studies that have presented different interventions as part of AMS to reduce the use of FQ in AUC. It would be useful to include these studies in the discussion with a comparison to the current study. Some examples; - Hayden DA, White BP, Neely S, Bennett KK. Impact of fluoroquinolones.

Response 6: Thank you very much for your valuable comment. We agree with this point. We have cited the study you provided, compared it with our own research. And we added the significance of our study (lines 232 to 238). Please kindly check it.

Reviewer 2 Report

Comments and Suggestions for Authors

This is a study on the efficacy of cephalosporins vs fluoroquinolones for the treatment of urinary tract infections.

The study and manuscript  have significant issues to be addressed.

First of all it ts unclear why levofloxacin was chosen over ciprofloxacin. Its bioavailability is not ideal for urinary tract infections and very importantly resistance over respiratory tract infections may be induced.

What are the exact resistance rates to levofloxacin? Is it not mentioned in the text. And how do the authors explain that the resistance rates are not decreased despite the reduced use of these drugs?

More analytical mention on the epidemiology of resistance of E coli from national data should be made in the manuscript.

What do the authors mean by treatment failure? Recurrence of the infection or need to change to other treatment? And if so which one? How is treatment success compatible with antimicrobial resistance?

Last but not least the significant difference in age between the two groups may indeed explain the advantage of cephalosporins which are not as efficacious in the case other male related urinary tract infections such as prostatitis.

The manuscript needs major revisions.

Comments on the Quality of English Language

Minor English editing is required

Author Response

Comments 1: First of all it ts unclear why levofloxacin was chosen over ciprofloxacin. Its bioavailability is not ideal for urinary tract infections and very importantly resistance over respiratory tract infections may be induced.

Response 1: Thank you very much for your valuable comment. We agree with this point. LVFX was likely preferred by prescribing physicians and patients because of its high bioavailability and once-daily dosing. In addition, ciprofloxacin is rarely used in Japan, have been discontinued, and is difficult to obtain. We added this to the discussion section (lines 183-187). Please kindly check it.

Comments 2: What are the exact resistance rates to levofloxacin? Is it not mentioned in the text. And how do the authors explain that the resistance rates are not decreased despite the reduced use of these drugs?

Response 2: Thank you very much for your valuable comment. We agree with this point. The resistance rate for levofloxacin is indicated by the antibiogram results. And outpatients who visit our hospital come from various regions. The detection status of drug-resistant bacteria in the community may be affected by the amount of antimicrobial drugs used in each region. Therefore, we inferred that the effect of reducing FQ usage in our hospital could not be confirmed in outpatient settings. We added this to the discussion section (lines 291-297). Please kindly check it.

Comments 3: More analytical mention on the epidemiology of resistance of E coli from national data should be made in the manuscript.

Response 3: Thank you very much for your valuable comment. We agree with this point. In the discussion section, we cited a reported Japanese surveillance study showing an increase in LVFX-resistant EC and ESBL-EC in women before and after menopause. We also cited that postmenopausal women have a higher risk of drug-resistant EC. In addition, we added that in this study, the proportion of microorganisms obtained from AUC patients and the proportion of ESBL-EC tended to be the same (lines 270-272). Please kindly check it.

Comments 4: What do the authors mean by treatment failure? Recurrence of the infection or need to change to other treatment? And if so which one? How is treatment success compatible with antimicrobial resistance?

Response 4: Thank you very much for your valuable comment. The definition of successful treatment is described in the Materials and Methods section (lines 381-386). Please kindly check it. In general, the bacteriological success of antibiotic treatment of infections is considered to be compatible with the results of antimicrobial susceptibility tests. In this study, we evaluated the relationship between the results of antibiograms and the efficacy of antimicrobials.

Comments 5: The significant difference in age between the two groups may indeed explain the advantage of cephalosporins which are not as efficacious in the case other male related urinary tract infections such as prostatitis.

Response 5: Thank you very much for your valuable comment. We agree with this point. As you pointed out, intravenous antimicrobials should be given as a priority for acute prostatitis. We have added a note regarding the usefulness of cephalosporins for male patients (lines 246-254). Please kindly check it.

Comments on the Quality of English Language Minor English editing is required: Thank you for your suggestion. We have reviewed the English wording and style of the entire text.

Round 2

Reviewer 1 Report

Comments and Suggestions for Authors

The authors completed the recommended revisions and answered the questions raised during the review.

Reviewer 2 Report

Comments and Suggestions for Authors

The authors have improved the manuscript sufficiently; it can now be published in its present form